# Tagitinin A regulates an F-box gene, CPR30, to resist tomato spotted wilt orthotospovirus (TSWV) infection in *Nicotiana benthamiana*

Jing Li[1�} ], Xiaoman Ai[1,2�} ], Suhua Zhang[2�} ], Xue Zheng[2], Lizhen Zhang[2], Jie Zhang[2]*, Lihua Zhao[2]*

1 College of Biological Science and Food Engineering, Southwest Forestry University, Kunming, China,
2 Yunnan Provincial Key Lab of Agricultural Biotechnology, Institute of Biotechnology and Germplasm Resources, Yunnan Academy of Agricultural Sciences, Kunming, China

☉ These authors contributed equally to this work.
* 470917665@qq.com (JZ); 356429686@qq.com (LZ)

**Data Availability Statement:** All relevant data are within the manuscript and its Supporting information files.

## Abstract

Tomato spotted wilt orthotospovirus (TSWV) is one of the most destructive pathogens and causes serious losses in agriculture worldwide. Biogenic pesticides application may be an effective approach for defending against TSWV. Tagitinin A (Tag A) extracted from *Tithonia diversifolia* (Hemsl.) A. Gray has a high protective effect against TSWV infection. Tag A can induce jasmonic acid to suppress gene expression in TSWV. In this study, the F-box protein (CPR30) was mediated by Tag A, the expression of the *CPR30* gene in Tag A-treated leaves was significantly higher (2 times) than that of the negative control. Furthermore, the replication of TSWV-*NSm/NSs* genes and the expression of TSWV-NSm/NSs proteins significantly increased after silencing the *CPR30* gene in protective assays; *CPR30* overexpression showed the opposite tendency. The CPR30 protein only localized and interacted with the TSWV-NSm protein. Thus, this study reveals a new mechanism by which Tag A mediates ubiquitin–protein ligase E3 (CPR30) to interact with NSm inhibit NSm replication and expression, and defend against systemic TSWV infection.

## Introduction

Tomato spotted wilt orthotospovirus (TSWV, synonym *Orthotospovirus tomatomaculae*) belongs to the species *Tospovirus*, genus *Orthotospovirus* and family Tospovirdae (ICTV, 2022), has more than 1000 host plants belonging to 85 families, and causes the largest yield loss in agriculture, ranking as the second most important plant viruses [1, 2]. TSWV consists of large (L), medium (M), and small (S) RNA genomes. M encodes the movement protein (NSm), and NSm plays a key role in TSWV movement and symptom development in the host [3]. S encodes the nucleocapsid protein (N) and the silencing suppressor protein (NSs), and the NSs protein is involved in TSWV replication and transmission processes [4, 5]. The N protein of TSWV interacts with genomic RNA to form viral ribonucleoprotein (vRNP) complexes that are functional templates for genomic RNA replication and viral mRNA transcription [6].

**Funding:** This work was supported by National Natural Science Foundation of China (No.32360639), Yunnan Young & Elite Talents Project (YNWR-QNRC-2020 to JZ, YNWR-QNRC-2022 to LHZ), Yunnan Provincial Natural Science Foundation (No. 202401AS070031). The funder of H.L.Z. conceived, designed the experiments, and wrote the manuscript, J.Z. analyzed the data and revised the manuscript.

**Competing interests:** The authors have declared that no competing interests exist.

Biogenic antiviral agents, whose effective components are compounds extracted from plants, offer a promising approach for pathogen control due to the absence of TSWV-resistant varieties and the environmentally unfriendly nature of insecticides. Previous studies have shown that biotic products are potential elicitors of induced resistance to pathogen infection. For example, eugenol is a plant-derived natural compound that can boost tomato plants' resistance to Tomato yellow leaf curl virus (TYLCV) by regulating the expression of the defense gene *SlPer1* [7]. As a signaling molecule, Benzothiadiazole (BTH) enhances the expression of defense genes (*PR1*, *PR4*) to inhibit Tobacco mosaic virus (TMV), Pepper golden mosaic virus (PepGMV), Zucchini yellow mosaic virus (ZYMV), and Cucumber mosaic virus (CMV) infection [8, 9]. Ferulic acid derivatives can significantly increase defense enzyme activity and induce secondary metabolite accumulation to inhibit TMV infection [10]. 3-Acetonyl-3-hydroxyoxindole (AHO), Reticine A, and oligosaccharides (COS) have been shown to effectively induce systemic acquired resistance (SAR) in tobacco and tomato through the salicylic acid (SA) to signaling pathway to defend against TSWV, TMV, CMV, and TYLCV infection [11–13]. However, the biotic products regulating ubiquitin to inhibit plant virus infection have not been reported.

Tagitinin A (Tag A) extracted from *Tithonia diversifolia* (Hemsl.) A. Gray has been reported to have pharmacological functions in medical treatments and biological activities in agriculture. Tag A inhibits carrageenan foot swelling in rats and treats diabetes through peroxisome proliferator-activated receptors (PPARs) [14, 15]. In agriculture, Tag A induces the jasmonic acid (JA) pathway and increases the activity of the defense enzyme phenylalanine ammonia-lyase (PAL) in *Nicotiana benthamiana* against TSWV and TMV infection [16–18]. In this study, we found that Tag A regulated the F-box protein (CPR30) interacting with the TSWV-NSm protein and inhibited TSWV infection. The present biogenic antiviral molecules will be helpful in the development of suitable eco-friendly formulations to mitigate TSWV infection in plants.

## Materials and methods

### Plant and virus material

*Nicotiana benthamiana* seeds were germinated in pots (500 mm height) filled with a mixture of humus (60%), perlite (30%), and roseite (10%). Plants were grown in an environmentally controlled greenhouse with a relative humidity of 60%, a 10/14-h dark/light photoperiod (100 μmol/m$^2$.s), and day/night temperatures of 25/28°C during the experimental period. At the 6-leaf stage, the plants were prepared for the experiments. Cultures of the virus (TSWV-KM isolates) were propagated continuously on *N. benthamiana* plants. Tag A derived from *T. diversifolia*, was provided by Professor Shunlin Li, KIB of CAS.

### RNA extraction and RT-qPCR analysis

Total RNA was isolated using an RNA-Easy™ Isolation Reagent (Vazyme, Nanjing, China). First-strand cDNA was synthesized from 1000 ng RNA using oligo-$^{dT}$ according to the manufacturer's instructions (HiScipt® III 1$^{st}$ Strand cDNA Synthesis Kit; Vazyme, Nanjing, China). RT-qPCR was performed using the Hieff® qPCR SYBR Green Master Mix System (YEASEN, Shanghai, China). The primers are presented in S1 Table. Reactions were performed in a final volume of 10 μL under the following thermal profile: 95°C for 10 min, followed by 40 cycles of 95°C for 10 s, 57°C for 30 s, and 72°C for 30 s.

Plasmid DNA was used to generate a standard curve for the gene sequences. The plasmid DNA was diluted for standard samples in a dilution series ($10^{-2}$, $10^{-3}$, $10^{-4}$, $10^{-5}$, and $10^{-6}$). Three replicates were included for each sample.

## Western blotting

The total protein was extracted from tobacco leaves (0.2 g, fresh weight) using TriPure Isolation Reagent (Vazyme, Nanjing, China) according to the manufacturer's instructions, and each sample containing the same quantity of extracted proteins weighed 10 mg. The proteins were boiled and denatured. An equal sample volume (10 μL) was loaded on a 12.5% polyacrylamide gel (10 × 10 cm), and proteins were separated by electrophoresis at 100 V for 90 min. The NSs and NSm proteins were detected using an anti-rabbit primary antibody (1:4000) and subsequently probed with AP-coupled goat anti-rabbit IgG (1:8000; Sigma, Santa Clara, USA). Three replicates were included for each sample.

## VIGS and overexpression assays for the *CPR30* gene

The 200-bp fragment of *CPR30* was amplified from *N. benthamiana* cDNA with specific primers (S1 Table) using Q5 Superfidelity DNA polymerase (NEB, English). The fragment was inserted into VIGS vector TRV-pTV00 to generate the constructed vector TRV-pTV00-CPR30 at restriction sites *BamHI* and *HindIII*. The constructed vector TRV-pTV00-CPR30 was transformed into *Agrobacterium tumefaciens* GV3101 and infiltrated into tobacco leaves. The plants of gene *CPR30* were silenced, and the protective assays were performed at 10–15 days post-inoculation (dpi). Tag A solution was spread onto whole leaves of *N. benthamiana* and after 6 h, TSWV was inoculated with the leaves that previously received the compound. The full-length *CPR30* sequence was amplified using specific primers (S1 Table) and ligated into the pCAMBIA1301-35 SN vector at restriction sites *BamHI* and *HindIII*. The recombinant plasmid was electro-transferred into *A. tumefaciens* GV3101. Plants overexpressing the *CPR30* gene were treated in the protective assays. For virus-induced gene silencing (VIGS) and overexpression assays, plants only infiltrated with the empty vector were used as the negative control, and plants only used in protective assays were positive controls, Three replicates were included for each sample.

## Protein interactions

The *CPR30* coding sequences were inserted into prey vector pGADT7, and NSs or NSm was introduced into bait vector pGBKT7. These vectors were then co-transformed into two-hybrid (Y2H) yeast competent cells using the Matchmaker Gold Yeast Two-Hybrid System (Clontech) for the Y2H assay.

The *NSm* and *NSs* coding sequences were fused with pSPY vectors containing the C-terminal cYFP fragment, and the *CPR30* sequence was fused with pSPY vectors containing the N-terminal nYFP. These expression constructs were then separately transformed into *Agrobacterium* strain GV3101 for transient expression in N. *benthamiana* leaves to perform the bimolecular fluorescence complementation (BiFC) assays. The NSm and NSs coding sequences were cloned into the nLUC vectors, and the *CPR30* coding sequences were cloned into cLUC. These expression constructs were then separately transformed into *Agrobacterium* strain GV3101 for transient expression in *N. benthamiana* leaves for split-luciferase (LUC) complementation assays. LUC activity was detected using the iBright 1500 plant imaging system (Thermo Fisher Scientific, USA). The experiments were repeated at least three times, and the primers are listed in S1 Table.

## Confocal laser scanning microscopy

The leaf epidermis was dissected from agroinfiltrated *N. benthamiana* leaves. GFP was captured at a wavelength of 488 nm, and emissions were captured at 497–520 nm. Moreover,

mCherry was captured at a wavelength of 561 nm and emissions were captured at 585–615 nm. YFP fluorescence was examined after 48 h with the inverted TCS SP8 and 10× water immersion objective lenses instrument. Images were processed using TCS SP8 and Adobe Photoshop (San Jose, CA, USA).

### Statistical analysis

One-way analysis of variance (ANOVA) was performed to compare significant differences among three or more factors. Significant differences were determined using SPSS (Version 17.0) at a significance level of $p < 0.05$, and the figures were generated using SigmaPlot version 10.0.

## Results

### The *CPR30* gene was regulated by Tag A

The expression levels of several genes, including F-box protein CPR30 (*QAt4g22390*), heptahelical transmembrane protein 4-like (*HTP-4, At4g37680*) involved in the AMPK signaling pathway, patatin-like protein 2 (*PLP2, LOC_Os08g37250*), ethylene-responsive transcription factor ERF017 (*At1g21910*) implicated in ethylene-activated signaling, transcription factor MYB48 (*At3g53200*), and kirola-like isoform X2 (*KIX2, At1g70830*) associated with pathogen defense responses, were examined in *Nicotiana tabacum* K326 leaves treated with Tag A [19–24]. The expression of the *CPR30* and *KIX2* genes was upregulated (Fig 1A and 1F), whereas only *CPR30* expression significantly increased in Tag A-treated leaves compared to positive control (TSWV-treated leaves) and mock control leaves (DMSO-treated leaves). The downregulated expression of genes including *HTP-4, PAP2, ERF017* and *MYB48* (Fig 1B–1E), which decreased in Tag A-treated leaves compared to positive and mock control leaves, was not

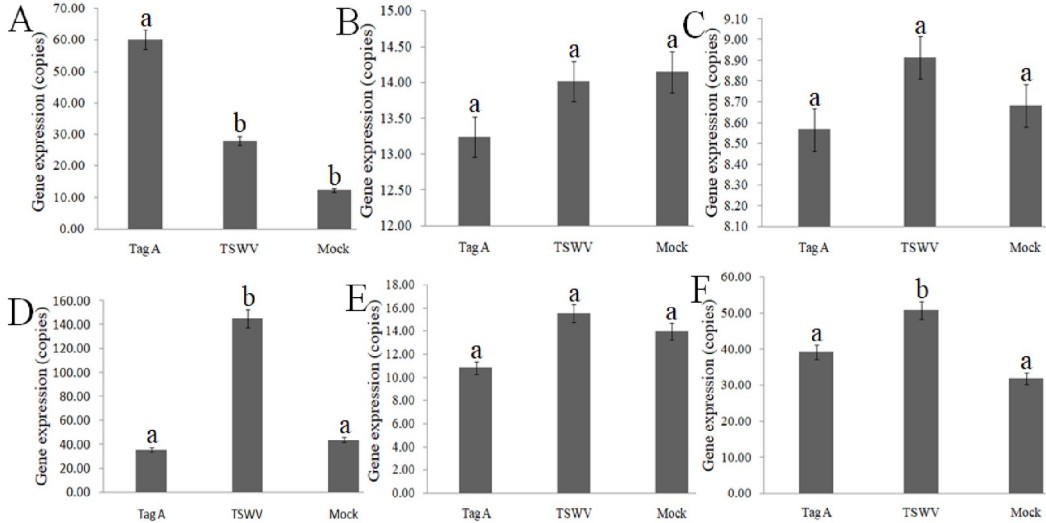

**Fig 1. The expression levels of genes were analyzed using RT-qPCR.** (A) The *CPR30* gene expressions in leaves of N. *benthamiana* treated with Tag A, TSWV or Mock were assessed. (B) The *HTP-4* gene expressions in N. *benthamiana* leaves with different treatment. (C) The *PLP2* gene expressions in N. *benthamiana* leaves with different treatment. (D) The *ERF17* gene expressions in N. *benthamiana* leaves with different treatment. (E) The *MYB48* gene expressions in N. *benthamiana* leaves with different treatment. (F) The *KIX2* gene expressions in N. *benthamiana* leaves with different treatment. Tag A: Tag A was smeared on leaves, TSWV: TSWV was inoculated on leaves, Mock: infiltrated with water for healthy plants. Significant differences among the three data sets are denoted by different letters (one-way ANOVAs, Tukey's HSD, $P < 0.05$).

significant. The expression levels of the *ERF017* and *KIX2* genes were significantly increased in TSWV-inoculated leaves compared to the negative control. These results suggest that the *CPR30* gene was induced by Tag A, further confirming its role in defense against TSWV infection.

## Tag A affected the expression of TSWV-*NSs*/*NSm* genes after silencing the *CPR30* gene

The expression levels of the *NSm*, and *NSs* genes were analyzed by RT-qPCR on inoculated and systemic leaves at 1–7 dpi after *CPR30* was silenced in the protective assays. The expression level of the *NSm* gene significantly increased at 2–6 dpi in the treated tobacco plants with inoculated and systemic leaves than in positive plant leaves (infiltrated with empty vector and treated in the protective assay). Gene expression peaked at 6 dpi (Fig 2A and 2B). The expression level of the *NSs* gene significantly increased at 2–6 dpi in treated tobacco plants with inoculated and systemic leaves compared to the positive plants. Gene expression peaked at 6 dpi in systemic tobacco plants (Fig 2C and 2D). The replication inhibition of *NSm* and *NSs* genes was weakened when the *CPR30* gene was silenced. Thus, the replication of *NSm* and *NSs* genes might be inhibited by CPR30, which was regulated by Tag A.

## Tag A affected the expression of TSWV genes after overexpression of the *CPR30* gene

The expression levels of the *NSm*, and *NSs* genes in the inoculated and systemic leaves at 1–7 dpi were analyzed by RT-qPCR in the protective assay after *CPR30* overexpression. The transcription levels of the *NSm* and *NSs* genes were significantly lower in inoculated and systemic

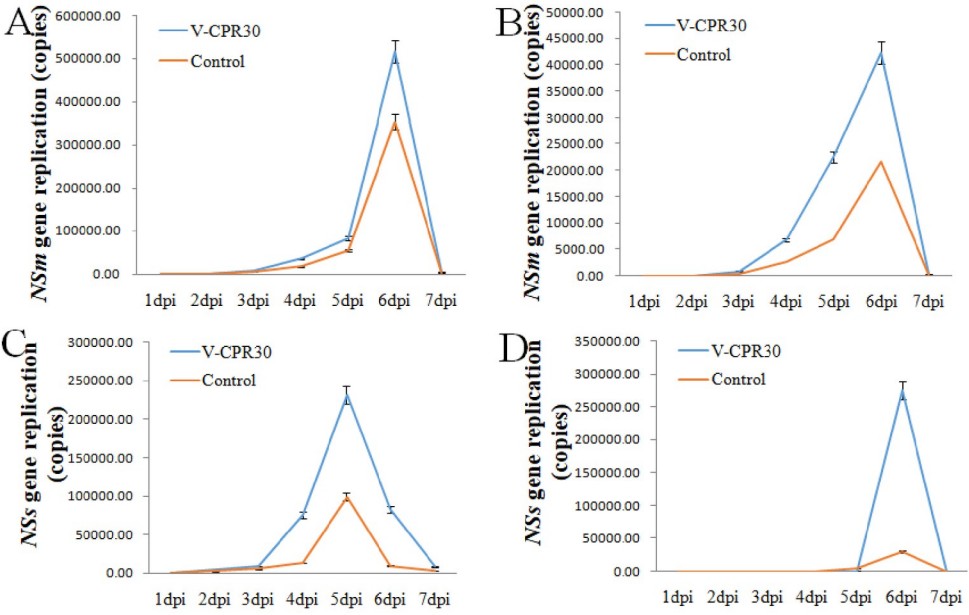

**Fig 2. The expression levels of TSWV genes after the *CPR30* gene silenced in the tobacco plants.** (A) TSWV *NSm* gene in the inoculated leaves after *CPR30* gene silenced. (B) TSWV *NSm* gene in the systematic leaves after *CPR30* gene silenced. (C) TSWV *NSs* gene in the inoculated leaves after *CPR30* gene silenced. (D) TSWV *NSs* gene in the systematic leaves after *CPR30* gene silenced. V- CPR30, the plants of *CPR30* gene silenced and treated with protective assay; Control, the plants of infiltrated empty vector and treated with protective assay.

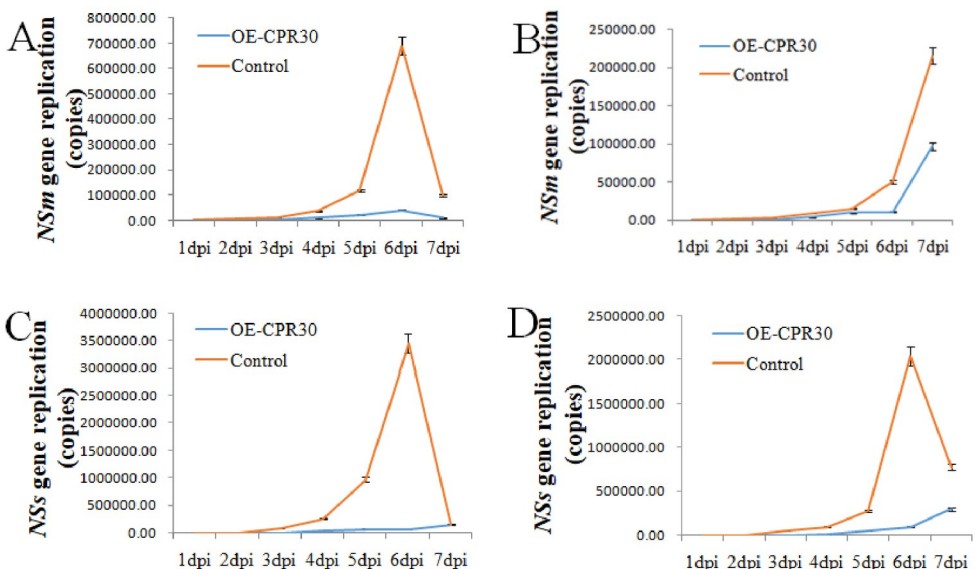

**Fig 3. The expression levels of TSWV genes in the *CPR30* gene overexpression plants.** (A) TSWV *NSm* gene expression in the inoculated leaves after *CPR30* gene overexpression. (B) TSWV *NSm* gene expression in the systematic leaves after *CPR30* gene overexpression. (C) TSWV *NSs* gene expression in the inoculated leaves after *CPR30* gene overexpression. (D) TSWV *NSs* gene expression in the systematic leaves after *CPR30* gene overexpression., OE-CPR30, the plants of *CPR30* gene over-expressed and treated with protective assay; Control, the plants of infiltrated empty vector and treated with protective assay.

*CPR30*-overexpressing tobacco plants than in the positive control plants (infiltrated with empty vector and treated in protective assays) at 3–7 dpi, and gene expression peaked at 6 dpi (Fig 3). The overexpression assays showed the opposite tendency in *CPR30*-silenced plants, further demonstrating that Tag A mainly regulates CPR30 to inhibit TSWV *NSm* and *NSs* gene expression.

## Tag A affected NSs and NSm protein expression after *CPR30* gene silencing and overexpression

The expression of both NSs and NSm was detected by Western blot at 5 dpi in the inoculated and systemic leaves from the protective assay after *CPR30* gene was silenced or overexpressed (Fig 4). The expression levels of the NSs and NSm proteins at 5 dpi in inoculated and systemic leaves of treated plants (*CPR30*-silenced plants treated in protective assays) were significantly higher than in those of the positive control (plants infiltrated with the empty vector and treated in protective assays) (Fig 4A). The expression levels of the NSm and NSs proteins at 5 dpi in inoculated and systemic leaves of treated plants (plants overexpressing the *CPR30* gene and treated in protective assays) were lower than those of the positive control (plants infiltrated with the empty vector and treated in protective assays) (Fig 4B). Taken together, the expression levels of the NSs and NSm protein were inhibited by Tag A through regulation of the *CPR30* gene in the protective assay.

## CPR30 interacted with the TSWV NSm protein

To further investigate whether NSs, NSm, and the CPR30 proteins were co-localized in living cells, we transiently expressed recombinant NSs-GFP, NSm-GFP, and CPR30-mCherry in leaf

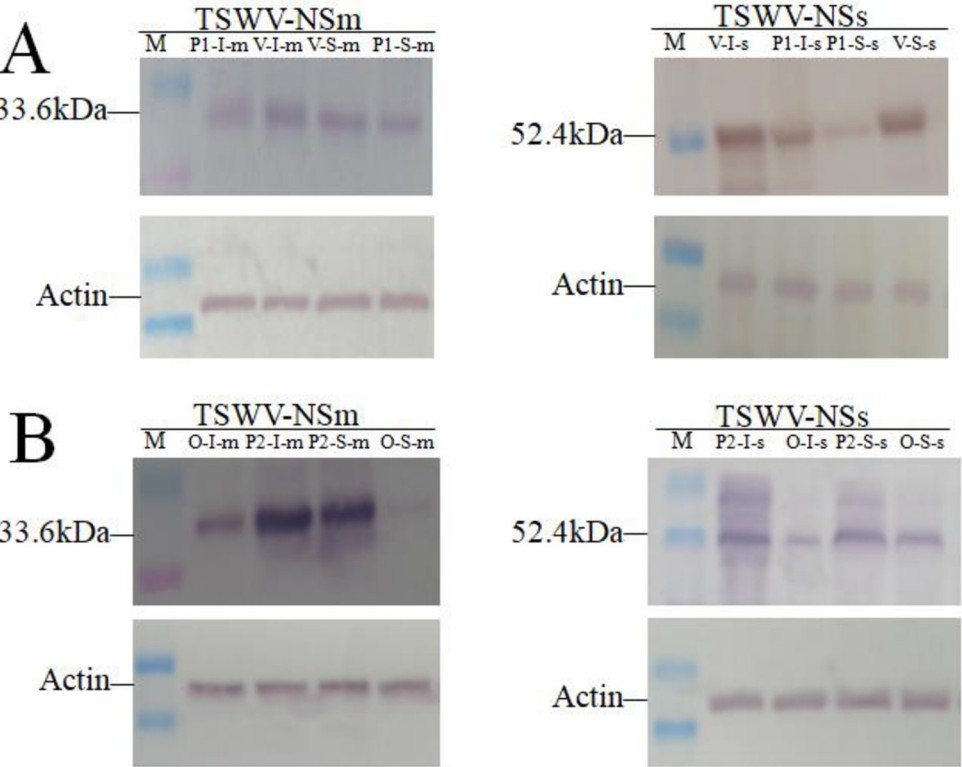

**Fig 4. The expression levels of TSWV proteins were analyzed.** (A) The expression of TSWV-NSm/NSs proteins on leaves of *CPR30* gene silenced and treated with protective assay. (B) The expression of TSWV-NSm/NSs proteins on leaves of *CPR30* gene over-expressed and treated with protective assay. M, Marker 250 kD; V: the plants of *CPR30* gene silenced and treated with protective assay; P1: the plants of infiltration empty vector and treated within protective assay; O: the plants of *CPR30* gene over-expressed and treated with protective assay; P2: the plants of infiltration empty vector and treated within protective assay; I: inoculated leaves; S: **systematic leaves; m:** TSWV-**NSm; s:** TSWV-**NSs; Actin:** RuBisco large subunit stained with R250 was used as a loading control.

epidermal cells of *N. benthamiana* by agroinfiltration. The co-expression of NSm-GFP and CPR30-mCherry was confirmed at the plasma membrane (Fig 5A).

Therefore, we examined these protein interactions under both in vitro and in vivo conditions. In BiFC assays, positive interactions indicated by yellow fluorescence were observed between CPR30-nYFP and NSm-cYFP in the cytoplasm and PM of leaf cells (Fig 5B). Furthermore, the LUC and Y2H assays validated the interactions between CPR30 and NSm (Fig 5C and 5D). These results confirmed that Tag A regulated the CPR30–NSm interaction to defend against TSWV infection.

## Discussion

Plant viruses, especially TSWV, can cause serious agricultural losses. At present, a promising approach to prevent or control diseases caused by plant viruses may be the use of botanical pesticides. Our previous studies showed that Tag A derived from T. *diversifolia* had antiviral activity against TSWV with both curative and protective effects and that Tag A defended against TSWV by activating the JA defense signaling pathway [17]. Based on our previous study, we selected other genes in the defense signaling pathway for RT-qPCR. The signaling

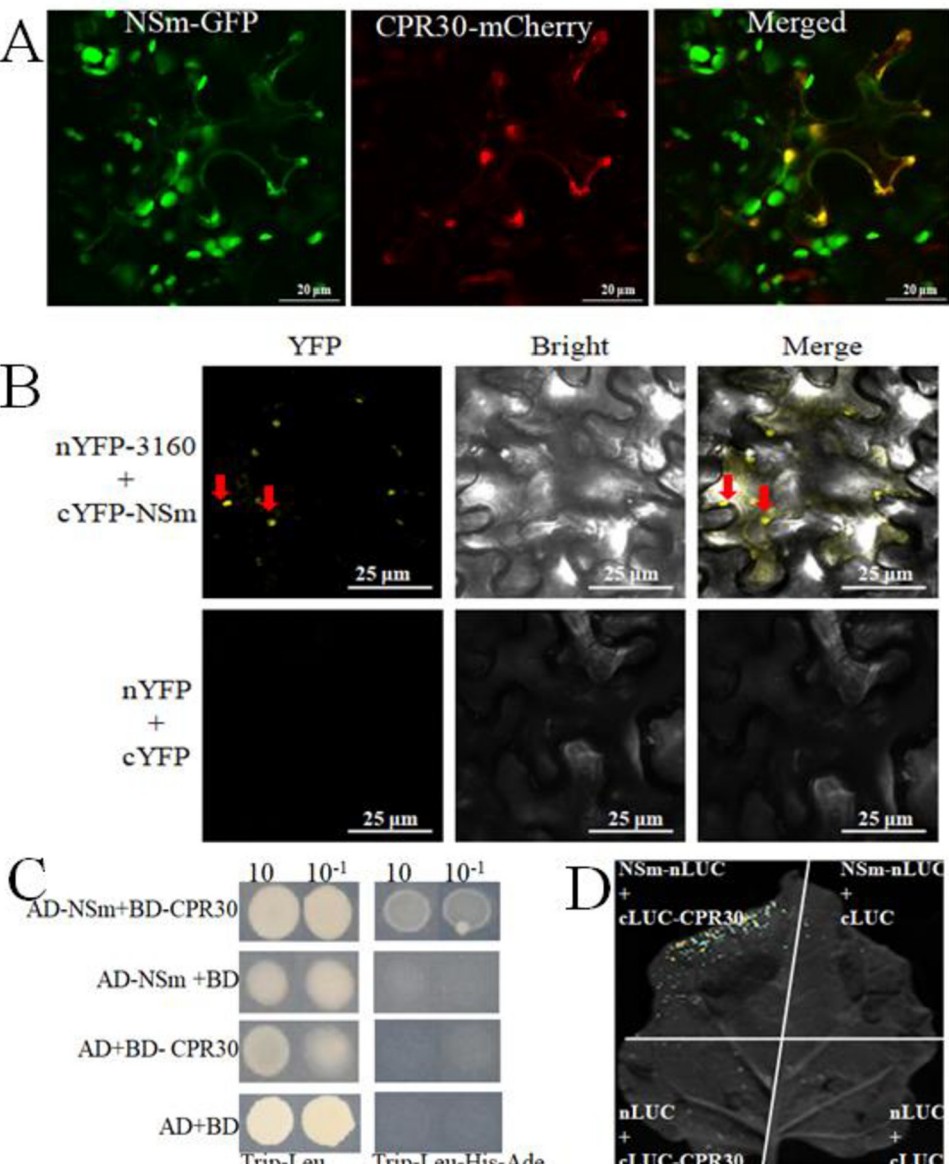

**Fig 5. The interaction of CPR30 and TSWV NSm protein.** (A) Co-localization of NSm-GFP and CPR30-mCherry intobacco leaves. (B) Bifc of CPR30-nYFP and NSm-cYFP. (C) Yeast two-hybrid experiment of CPR30 and TSWV-NSm, the combinations with the empty vectors pGADT7(AD) or pGBKT7(BD) were used as negative control. (D) LUC of CPR30-cYFP and NSm-nYFP. Ade, His, Leu, and Trp: Interactions between bait and preywere determined by cell growth on synthetic dropout medium lacking Ade, His, Leu, and Trp.

pathways of AMPK, ET, SA and transcription factor MYB48 were not induced by Tag A; however, a component of ubiquitin–protein ligase E3 (CPR30) was induced by Tag A.

Biogenic antiviral agents usually resist viral infection by inducing potential elicitors in the host [11, 13]. In this study, we observed that the expression of both NSs and NSm proteins from TSWV was inhibited by the CPR30 protein, whose expression was regulated by Tag A. Furthermore, our findings suggest that CPR30 directly interacts with the NSm protein, thereby interfering with virus movement. CPR30 is an ubiquitin-protein ligase E3 and encodes a

functional F-box protein that interacts with multiple Arabidopsis-SKP1-like (ASK) and SNC1 proteins to regulate plant immunity [22, 25]. *CPR30* is allelic to *CPR1* and involved in complex-mediated stability control of plant NLR proteins (SNC1 and RPS2) in regulating their protein levels and prevent autoimmunity [26]. The *CPR30* gene induces SAR to defend against *Verticillium* wilt in *G. hirsutum* [27]. This is the first report showing that CPR30 is regulated by Tag A and interacts with virus proteins to resist virus infection. Biogenic antiviral agents usually have multiple antiviral mechanisms, including the inhibition of viral genome replication, movement, and assembly and the induction of plant resistance against viruses. Some botanical products can directly inhibit the replication virus in plants. Gramine analogues can significantly interfere with the assembly of TMV rods by cross-linking with the capsid protein [28]. Some botanical products induce plant resistance through JA signaling or resistance-related genes and proteins. Previous studies have shown that ningnanmycin (a derivative of the marine natural product essramycin), Tagitinin C (Ses-2), and 1β-methoxydiversifolin-3-0-methyl ether (Ses-5) promote the accumulation of pathogen-related proteins (PRs) to prevent replication of the *cp* and *RdRp* genes, which comprise the viral assembly of TMV [16, 29, 30]. In this study, we proposed that Tag A application on the host induced the host defense responses gene *CPR30*, and CPR30 interacted with NSm to prevent TSWV infection. This is the first report showing that the CPR30 protein, regulated by Tag A, co-localized and directly interacted with TSWV-NSm. CPR30 is a negative factor that regulates R proteins and degrades the positive regulator of plant defense through the 26S proteasome [22]. Here, CPR30 was revealed as a positive regulator that directly interacts with NSm to defend against TSWV infection, providing a new theory for the effective prevention and control of TSWV.

## Conclusions

This study showed that *CPR30* gene expression was two times higher in Tag A-treated leaves than in the negative control. Moreover, the transcription levels of the *NSm* and *NSs* genes and the expression of the NSm and NSs proteins were significantly higher in TRV-*CPR30* and significantly lower in *CPR30*-overexpressing tobacco plants than in the inoculated and systemic leaves of the positive control. Further research showed that CPR30 co-localized and directly interacted with TSWV-NSm. Taken together, our results indicated that Tag A activated CPR30 to induce systemic resistance, inhibiting TSWV gene replication and protein expression and defending against TSWV infection. The results indicated the unique mechanism by which Tag A regulated a plant defense response gene to defend against TSWV infection. Future studies should evaluate this as a tool for developing virus-resistant varieties.

## Supporting information

**S1 Raw images.**
(PDF)

**S1 Table. The primers used in this.**
(DOCX)

## Acknowledgments

Authors would like to thank Jianqiang Wu Dr for providing the vectors for VIGS.

## Author Contributions

**Conceptualization:** Lihua Zhao.

**Data curation:** Xue Zheng, Lizhen Zhang.

**Formal analysis:** Jing Li.

**Funding acquisition:** Jie Zhang, Lihua Zhao.

**Methodology:** Xiaoman Ai, Suhua Zhang, Lizhen Zhang.

**Resources:** Xue Zheng.

**Validation:** Jing Li, Xue Zheng.

**Writing – original draft:** Jing Li, Xiaoman Ai.

**Writing – review & editing:** Jie Zhang, Lihua Zhao.

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
