## [Decision Letter · Decision Letter 0]

11 Sep 2024

PONE-D-24-36078Tagitinin A regulating an F-box gene, CPR30 to resist tomato spotted wilt orthotospovirus infectionPLOS ONE

Dear Dr. Zhao,

Thank you for submitting your manuscript to PLOS ONE. After careful consideration, we feel that it has merit but does not fully meet PLOS ONE’s publication criteria as it currently stands. Therefore, we invite you to submit a revised version of the manuscript that addresses the points raised during the review process. Please submit your revised manuscript by Oct 26 2024 11:59PM. If you will need more time than this to complete your revisions, please reply to this message or contact the journal office at plosone@plos.org. Please include the following items when submitting your revised manuscript:A rebuttal letter that responds to each point raised by the academic editor and reviewer(s). You should upload this letter as a separate file labeled 'Response to Reviewers'.A marked-up copy of your manuscript that highlights changes made to the original version. You should upload this as a separate file labeled 'Revised Manuscript with Track Changes'.An unmarked version of your revised paper without tracked changes. You should upload this as a separate file labeled 'Manuscript'.

We look forward to receiving your revised manuscript.

Kind regards,

Sumit Jangra, Ph.D.

Academic Editor

PLOS ONE

Journal Requirements:

1. When submitting your revision, we need you to address these additional requirements. Please ensure that your manuscript meets PLOS ONE's style requirements, including those for file naming. The PLOS ONE style templates can be found at https://journals.plos.org/plosone/s/file?id=wjVg/PLOSOne_formatting_sample_main_body.pdf and https://journals.plos.org/plosone/s/file?id=ba62/PLOSOne_formatting_sample_title_authors_affiliations.pdf 2. Thank you for stating the following financial disclosure: "This work was carried out with supporting from National Natural Science Foundation of China (No.32360639), Yunnan Young & Elite Talents Project (YNWR-QNRC-2020 to JZ, YNWR-QNRC-2022 to LHZ), Yunnan Provincial Natural Science Foundation (No. 202401AS070031)." Please state what role the funders took in the study.  If the funders had no role, please state: "The funders had no role in study design, data collection and analysis, decision to publish, or preparation of the manuscript." If this statement is not correct you must amend it as needed. Please include this amended Role of Funder statement in your cover letter; we will change the online submission form on your behalf. 3. Thank you for stating the following in the Acknowledgments Section of your manuscript: "This work was carried out with supporting from National Natural Science Foundation of China (No.32360639), Yunnan Young & Elite Talents Project (YNWR-QNRC-2020 to JZ, YNWR-QNRC-2022 to LHZ), Yunnan Provincial Natural Science Foundation (No. 202401AS070031)." We note that you have provided funding information that is not currently declared in your Funding Statement. However, funding information should not appear in the Acknowledgments section or other areas of your manuscript. We will only publish funding information present in the Funding Statement section of the online submission form. Please remove any funding-related text from the manuscript and let us know how you would like to update your Funding Statement. Currently, your Funding Statement reads as follows: "This work was carried out with supporting from National Natural Science Foundation of China (No.32360639), Yunnan Young & Elite Talents Project (YNWR-QNRC-2020 to JZ, YNWR-QNRC-2022 to LHZ), Yunnan Provincial Natural Science Foundation (No. 202401AS070031)." Please include your amended statements within your cover letter; we will change the online submission form on your behalf. 4. Please provide a complete Data Availability Statement in the submission form, ensuring you include all necessary access information or a reason for why you are unable to make your data freely accessible. If your research concerns only data provided within your submission, please write "All data are in the manuscript and/or supporting information files" as your Data Availability Statement. 5. PLOS ONE now requires that authors provide the original uncropped and unadjusted images underlying all blot or gel results reported in a submission’s figures or Supporting Information files. This policy and the journal’s other requirements for blot/gel reporting and figure preparation are described in detail at https://journals.plos.org/plosone/s/figures#loc-blot-and-gel-reporting-requirements and https://journals.plos.org/plosone/s/figures#loc-preparing-figures-from-image-files. When you submit your revised manuscript, please ensure that your figures adhere fully to these guidelines and provide the original underlying images for all blot or gel data reported in your submission. See the following link for instructions on providing the original image data: https://journals.plos.org/plosone/s/figures#loc-original-images-for-blots-and-gels. In your cover letter, please note whether your blot/gel image data are in Supporting Information or posted at a public data repository, provide the repository URL if relevant, and provide specific details as to which raw blot/gel images, if any, are not available. Email us at plosone@plos.org if you have any questions.

Reviewers' comments:

Reviewer's Responses to Questions

**Comments to the Author**

1. Is the manuscript technically sound, and do the data support the conclusions?

Reviewer #1: Yes

Reviewer #2: Partly

Reviewer #3: Partly

Reviewer #4: Yes

2. Has the statistical analysis been performed appropriately and rigorously? 

Reviewer #1: N/A

Reviewer #2: Yes

Reviewer #3: Yes

Reviewer #4: N/A

3. Have the authors made all data underlying the findings in their manuscript fully available?

Reviewer #1: No

Reviewer #2: Yes

Reviewer #3: Yes

Reviewer #4: Yes

4. Is the manuscript presented in an intelligible fashion and written in standard English?

Reviewer #1: No

Reviewer #2: No

Reviewer #3: Yes

Reviewer #4: Yes

5. Review Comments to the Author

Reviewer #1: Dear Authors,

The findings of the manuscript displays a route map to exploit metabolites of plant origin to mitigate viral diseases. This will provide knowledge on tritrophic interactions. But the paper is with lot of grammar mistakes and English is very poor. The paper can be readable only if it is modified with good and correct English.

Few comments as sample:

Title has to be changed as “Tagitinin A regulating an F-box gene, CPR30 to resist Orthotospovirus tomatomaculae (TSWV) infection

The paper describes all studies only in N.bethamiana as model, in that case N.benthamana must come in title???

Line 33: Give the new name of Tomato spotted wilt orthotospovirus as “Orthotospovirus tomatomaculae, which is a new name assigned, please refer ICTV 2023 (TSWV) belongs to the species Tospovirus and add ICTV link as reference. Then give the old name TSWV inside the manuscript.

Line 57-58- Modify the sentence “resistance systemic acquired resistance (SAR)” to get correct meaning

Line 60: Change as “However, the biotic products regulating

68-69: Grammar mistake. Modify the sentence.

Reviewer #2: Previously, the authors found that Tagitinin A (Tag A) can suppress the expression of NSm (the movement protein) and NSs (the silencing suppressor protein) of tomato spotted wilt virus (TSWV). In this study, the authors investigated the molecular mechanism by which Tag A combats the TSWV using the yeast two-hybrid (Y2H) and virus-induced gene silencing (VIGS) assays. The results showed that TSWV-NSm can induce the expression of the F-box protein (CPR30) gene and interact with its translational product. Silencing of the CPR30 could significantly increase the replication and expression of TSWV-NSm/NSs, whereas over-expression of CPR30 displayed contrary results. These findings show a new mechanism in Tag A mediating CPR30, which mainly acts on NSm protein to prevent TSWV systematic infection. However, the present manuscript is deficient in several respects, including deficiencies in English language proficiency and inaccuracies in text formatting. Therefore, I recommend a Major Revision of the manuscript. Some comments are given below.

1. The first major issue with the manuscript is the quality of the English. On first reviewing the paper, I made numerous edits/changes to the text and found myself unnecessarily distracted from assessing the paper on scientific grounds. I appreciate that English may not be the first language of the authors, but there is no doubt that the quality of the paper will be affected.

2. It is recommended that the authors rewrite the sections on Materials and methods and Results to improve the logical coherence between the individual methods and results.

3. The visual representation presented in Figure 4 is not sufficiently clear.

4. The authors should follow the PLOS ONE Guidelines for Authors and ensure that precise use of punctuation, font, size, line spacing, and paragraph formatting is followed.

5. Please check all the references and make sure that they conform to acceptable formats of PLOS ONE.

Reviewer #3: In this manuscript, the authors identified the effects of biogenic pesticide TagA on tomato spotted wilt orthotospovirus （TSWV）by regulating an F-box gene, CPR30. They indicated that the expression of the CPR30 gene in Tag A-treated leaves was significant higher (2 times) than that the negative control. And the replication of TSWV-NSm/NSs genes and the expression of TSWV-NSm/NSs proteins significantly increased after the CPR30 gene

was silenced in protective assays, over-expression of CPR30 showed the contrary tendency. the CPR30 protein only localizes and interacts with TSWV-NSm protein. Overall, this study directly demonstrates the regulatory function of CPR30 during tomato spotted wilt orthotospovirus infection. Otherwise, in current manuscript, about the real relationship among Tag A, CPR30 and resisting TSWV infection may less direct or clear. If the authors have already conducted the TagA treatment with CPR30 expression level (up and down-regulated) together at the same time, to compare the effects of TagA treatment, CPR30 expression level and TagA+CPR30 expression level on spotted wilt orthotospovirus infection, it is suggested to supplement the treatments and the results clear to the manuscript to clarify which is the primary activator and how the reaction each other to spotted wilt orthotospovirus infection.

Reviewer #4: The manuscript is well written but followings point will enhance the quality of manuscript.

Abstract lack novel statement that how it is different from previous research.

Materials and methods should be brief.

There should be need for the project in introduction showing that how this research is needed in the field of plant pathology.

Discussion should be expanded.

Please conclude your research and elaborate the future prospect.

6. PLOS authors have the option to publish the peer review history of their article (what does this mean?). If published, this will include your full peer review and any attached files.

Reviewer #1: No

Reviewer #2: No

Reviewer #3: No

Reviewer #4: **Yes: **Yasir Iftikhar

---

## [Author Response · Author response to Decision Letter 0]

25 Oct 2024

We are grateful to the reviewers and editor for their valuable comments on our paper, which helped us improving our manuscript significantly. We have done a thorough revision and believe that the revised version has been substantially improved accordingly. Enclosed please find our point-by-point responses to all the comments and suggestions from the reviewers and editor.Thank you for your consideration and time. We hope that you will find our revised manuscript satisfactory and look forward to having your further editorial contact.

---

## [Decision Letter · Decision Letter 1]

25 Nov 2024

Tagitinin A regulates an F-box gene, CPR30, to resist tomato spotted wilt orthotospovirus (TSWV) infection in Nicotiana benthamiana

PONE-D-24-36078R1

Dear Dr. Zhao,

We’re pleased to inform you that your manuscript has been judged scientifically suitable for publication and will be formally accepted for publication once it meets all outstanding technical requirements.

Kind regards,

Sumit Jangra, Ph.D.

Academic Editor

PLOS ONE

Additional Editor Comments (optional):

Reviewers' comments:

Reviewer's Responses to Questions

**Comments to the Author**

1. If the authors have adequately addressed your comments raised in a previous round of review and you feel that this manuscript is now acceptable for publication, you may indicate that here to bypass the “Comments to the Author” section, enter your conflict of interest statement in the “Confidential to Editor” section, and submit your "Accept" recommendation.

Reviewer #2: All comments have been addressed

Reviewer #4: All comments have been addressed

2. Is the manuscript technically sound, and do the data support the conclusions?

Reviewer #2: Yes

Reviewer #4: Yes

3. Has the statistical analysis been performed appropriately and rigorously? 

Reviewer #2: Yes

Reviewer #4: Yes

4. Have the authors made all data underlying the findings in their manuscript fully available?

Reviewer #2: Yes

Reviewer #4: Yes

5. Is the manuscript presented in an intelligible fashion and written in standard English?

Reviewer #2: Yes

Reviewer #4: Yes

6. Review Comments to the Author

Reviewer #2: (No Response)

Reviewer #4: (No Response)

7. PLOS authors have the option to publish the peer review history of their article (what does this mean?). If published, this will include your full peer review and any attached files.

Reviewer #2: No

Reviewer #4: **Yes: **Yasir Iftikhar

---

## [Editor Report · Acceptance letter]

29 Nov 2024

PONE-D-24-36078R1 

PLOS ONE

Dear Dr. Zhao, 

I'm pleased to inform you that your manuscript has been deemed suitable for publication in PLOS ONE. Congratulations! Your manuscript is now being handed over to our production team.

Kind regards, 

on behalf of

Dr. Sumit Jangra 

Academic Editor

PLOS ONE